# Spatial close-kin mark-recapture methods to estimate dispersal parameters and barrier strength for mosquitoes

John M. Marshall[1,2]*, Shuyi Yang[1], Jared B. Bennett[1],
Igor Filipović[3], Gordana Rašić[3]

1 Divisions of Biostatistics and Epidemiology, School of Public Health, University of California, Berkeley, California, United States of America, 2 Innovative Genomics Institute, Berkeley, California, United States of America, 3 Mosquito Genomics, QIMR Berghofer Medical Research Institute, Brisbane, Queensland, Australia

* john.marshall@berkeley.edu

## Abstract

Close-kin mark-recapture (CKMR) methods have recently been used to infer demographic parameters for several aquatic and terrestrial species. For mosquitoes, the spatial distribution of close-kin pairs has been used to estimate mean dispersal distance, of relevance to vector-borne disease transmission and genetic biocontrol strategies. Close-kin methods have advantages over traditional mark-release-recapture (MRR) methods as the mark is genetic, removing the need for physical marking and recapturing that may interfere with movement behavior. Here, we extend CKMR methods to accommodate spatial structure alongside life history for mosquitoes and comparable insects. We derive kinship probabilities for parent-offspring and full-sibling pairs in a spatial context, where an individual in each pair may be a larva or adult. Using the dengue vector *Aedes aegypti* as a case study, we use an individual-based model of mosquito life history to test the effectiveness of this approach at estimating parameters such as mean dispersal distance, daily staying probability, and the strength of a barrier to movement. Considering a simulated population of 9,025 adult mosquitoes arranged on a 19-by-19 grid, we find the CKMR approach provides unbiased and precise estimates of mean dispersal distance given a total of 2,500 adult females sampled over a three-month period using 25 traps evenly spread throughout the landscape. The CKMR approach is also able to estimate parameters of more complex dispersal kernels, such as the daily staying probability of a zero-inflated exponential kernel, or the strength of a barrier to movement, provided the magnitude of these parameters is greater than 0.5. These results suggest that CKMR provides an insightful characterization of mosquito dispersal that is complementary to conventional MRR methods.

**Data availability statement:** The source code for the individual-based mosquito simulation model is available at https://github.com/GilChrist19/mPlex. Documentation, including vignettes, are included for all simulation functions. The source code for inferring parameters based on the likelihood of the kinship data is available at https://github.com/MarshallLab/CKMR. Both sets of code are available under the GPL3 License and are free for other groups to modify and extend as needed.

**Funding:** This work was supported by a National Institutes of Health R01 grant (1R01AI143698) awarded to JMM and GR, and by a National Institutes of Health R01 grant (1R01AI190001) and funds from the Gates Foundation (INV-078535) awarded to JMM. The funders had no role in the study design, data collection and analysis, decision to publish, or preparation of the manuscript.

**Competing interests:** The authors have declared that no competing interests exist.

## Author summary

Close-kin mark-recapture (CKMR) is a genetic analogue of mark-release-recapture (MRR) in which the frequency of genetically-inferred familial relationships in a sample is used to infer demographic parameters such as census population size and mean dispersal distance. These methods have been widely applied to aquatic species; however their application to mosquitoes is yet to be rigorously explored. Previous theoretical work demonstrated the potential for CKMR to infer parameters such as population size and mortality rate for randomly-mixing mosquito populations, and close-kin-based methods have been used to infer movement patterns for *Aedes aegypti* mosquitoes in Singapore and Malaysia. Here, we use simulations to explore the potential for formal CKMR methods to characterize mosquito dispersal patterns. We find that formal CKMR methods are able to accurately estimate mean dispersal distance, and to estimate additional parameters, such as the strength of a landscape barrier and the probability that a mosquito remains within its population node each day. CKMR and other close-kin-based methods provide insights into mosquito dispersal complementary to commonly-used alternatives such as MRR, as they capture displacement across several generations and are not compromised by the marking process.

## 1 Introduction

Malaria, dengue, chikungunya and other mosquito-borne diseases continue to pose a major burden throughout much of the world [1,2]. Novel biological and genetics-based interventions, such as releases of mosquitoes infected with *Wolbachia* or engineered with gene drives, offer much promise to complement traditional control tools such as insecticide-treated nets, vaccines and antimalarial drugs. A common feature of these novel tools is the need for a detailed understanding of mosquito movement in order to design effective field trials and interventions, and to address biosafety concerns. In a recent randomized controlled trial (RCT) of *Wolbachia*-based population replacement of the dengue vector, *Aedes aegypti*, in Yogyakarta, Indonesia, *Wolbachia* was observed to spread significantly from intervention to control areas within one year of release [3]. This highlights the importance of quantifying mosquito movement to determine optimal spatial units for vector control RCTs. Predicting intentional geographic spread of self-propagating interventions such as *Wolbachia* and gene drive is also crucial, as is assessing the potential for confinement and logistics of reversibility during a trial [4].

A handful of methods are available to characterize mosquito movement patterns. The most direct of these is mark-release-recapture (MRR), hundreds of which studies have been conducted for *Ae. aegypti* and the malaria vector *Anopheles gambiae* in recent decades [5]. In MRR, a portion of a population is captured, marked and released, and subsequent collections are checked for recaptures. The fraction of recaptures over time can be used to infer population size and daily mortality, while times and distances between release and recapture events can be used to infer dispersal patterns. A major shortcoming of MRR is that inferred dispersal patterns

may be modified by the process of marking and capturing, and for mosquitoes specifically, releasing females may increase the risk of local disease transmission. Several genetic methods are available to characterize mosquito movement on a larger spatial scale, effectively estimating dispersal averaged over several generations. Wright's fixation index, $F_{ST}$, can be calculated using genetic markers such as single nucleotide polymorphisms or microsatellites [6,7], and population assignment tests can be used to infer movement between well-structured populations at a scale beyond the mean dispersal range of a species [8].

Close-kin mark-recapture (CKMR) is a promising new approach with the potential to complement these methods and deepen understanding of mosquito dispersal. In CKMR, the detection of a close-kin pair (parent-offspring, siblings, etc.) in a sample is analogous to the recapture of a marked individual in the MRR method [9]. Detection of several close-kin pairs separated by a given distance provides information about movement that occurred over a small number of generations, thus informing dispersal patterns on an intermediate spatial scale, between that of MRR methods and most other genetic approaches. Advantages of CKMR methods stem from the mark being a genetically-inferred close-kin relationship, removing the need for physical marking and recapturing. Three recent studies have used the spatial distribution of close-kin pairs to characterize *Ae. aegypti* and *Aedes albopictus* movement patterns [10–12], all using approaches inspired by the CKMR formalism [9]. Jasper et al. [10] estimate relatedness across three orders of kinship and estimate life stage-specific dispersal by considering possible life history events between each kinship pair. Filipović et al. [11] consider up to three orders of kinship and use coordinates of close-kin pairs to infer the distribution of distances between the birth and ovipositing sites of breeding females, and fit these distances to a variety of dispersal kernels. Ontiveros et al. [12] use genetic and network analyses to show *Ae. albopictus* act as a well-mixed population within an urban area, and use a refined version of the method of Filipović et al. [11] to fit a dispersal kernel.

Here, we develop and apply formal CKMR methods with spatial structure to estimate movement parameters of *Ae. aegypti*. Formal CKMR methods are based on explicit calculations of kinship probabilities, and the likelihood of observing a given number and category of close-kin pairs across space and time [9]. While the availability of spatial kinship information has been discussed as having the potential to inform dispersal parameters within a formal CKMR framework [9,13,14], very few studies have explored this through simulation [15–17]. Our approach builds upon CKMR methods specific to the life history of mosquitoes, previously used to estimate demographic parameters such as census population size, and adult and larval mortality rates [18]. As in that study, we use an *in silico* model of mosquito life history, this time with spatial structure, to generate kinship data and validate our inference methods. Using this approach, we determine optimal sampling schemes (sampled life stages, sample size and spatial distribution of traps) to accurately and efficiently estimate dispersal parameters, including mean dispersal distance, daily staying probability, and the strength of a barrier to movement.

## 2 Methods

### 2.1 Mosquito life history and spatial structure

As per our previous mosquito CKMR study [18], we base our analysis on a discrete-time version of the lumped age-class model [19,20], applied to mosquitoes [21] (Fig 1A). This model considers discrete life history stages - egg (E), larva (L), pupa (P) and adult (A) - with sub-adult stages having defined durations - $T_E$, $T_L$ and $T_P$ for eggs, larvae and pupae, respectively. We use a daily time-step, since mosquito samples tend to be recorded by day, and this is adequate to model the organism's population dynamics [22]. Daily mortality rates vary according to life stage - $\mu_E$, $\mu_L$, $\mu_P$ and $\mu_A$ for eggs, larvae, pupae and adults, respectively - and density-dependent mortality occurs at the larval stage. Sex is modeled at the adult stage - half of pupae emerge as females (F), and the other half as males (M). Given the rareness of female re-mating for important mosquito vector species (<1% for *An. gambiae* [23] and <1-5% for *Ae. aegypti* [24]), we assume that females mate once upon emergence, and retain the genetic material from that mating event for the remainder of

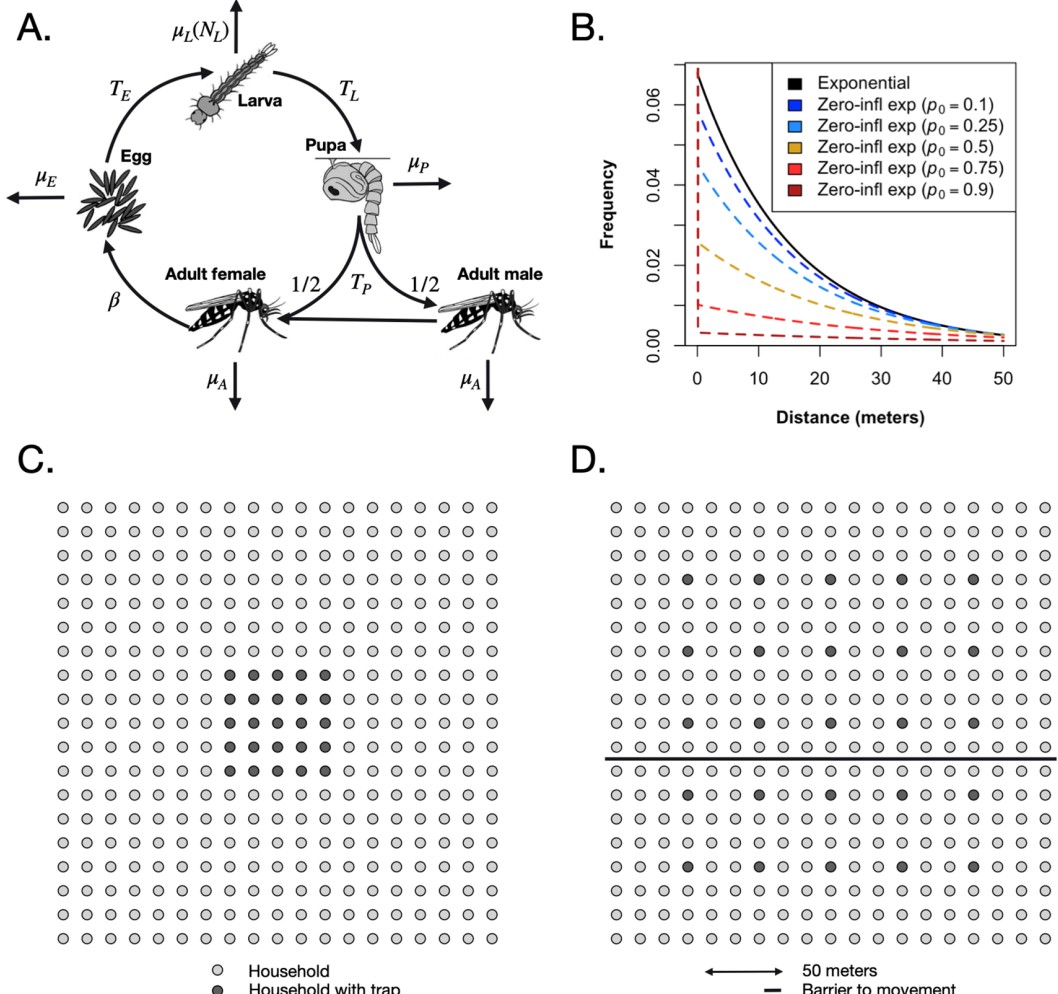

**Fig 1. Mosquito life history and spatial structure.** In the lumped age-class model, mosquitoes are divided into four life stages: egg, larva, pupa and adult **(A)**. The durations of the sub-adult stages are $T_E$, $T_L$ and $T_P$ for eggs, larvae and pupae, respectively. Sex is modeled at the adult stage, with half of pupae developing into females and half developing into males. Daily mortality rates vary by life stage - $\mu_E$, $\mu_L$, $\mu_P$ and $\mu_A$ for eggs, larvae, pupae and adults, respectively. Density-dependent mortality occurs at the larval stage and is a function of the total number of larvae, $N_L$. Females mate once upon emergence, and retain the genetic material from that mating event for the remainder of their lives. Males mate at a rate equal to the female emergence rate. Females lay eggs at a rate, $\beta$. In the spatial extension of the lumped age-class model, mosquito populations are distributed in space, with movement between them defined by an exponential (solid line) or zero-inflated exponential dispersal kernel (dashed lines) **(B)**. The daily probability of remaining in the same population, $p_0$, is varied while preserving the mean dispersal distance. This value is trimmed from the plot, but specified in the key. Mosquito populations are distributed according to a 19-by-19 grid of households (circles), with mosquito traps distributed in select households (black circles) according to the sampling scheme **(C)**. In some simulations and analyses, a barrier to movement is included (solid line) **(D)**.

their lives. Males mate at a rate equal to the female emergence rate which, for a population at equilibrium, is equal to the female mortality rate, $\mu_A$. Females lay eggs at a rate, $\beta$, which is assumed to be independent of age.

In extending the lumped age-class model to space, we consider a spatial distribution of mosquito populations, with movement between them defined by a dispersal kernel (Fig 1B–1D). Discrete populations in the resulting metapopulation are considered to be randomly mixing populations to which the lumped age-class model applies. The resolution of the individual populations (in terms of size) should be chosen according to the dispersal properties of the species being considered. For *Ae. aegypti*, populations on the scale of households may be appropriate, as this species tends to disperse

PLOS Computational Biology

locally, often remaining in the same household for the duration of their lifespan [25]. For *An. gambiae*, dispersal occurs over larger distances and villages may be an appropriate population unit [6]. By default in this paper, daily per-capita movement probabilities between populations are derived from an exponential dispersal kernel (Fig 1B). For populations $i$ and $j$ a distance $d_{ij}$ apart, the rate of movement between them is:

$$m_{ij} = \frac{\exp(-\lambda_d d_{ij})}{\sum_{j=1}^{n} \exp(-\lambda_d d_{ij})}. \tag{1}$$

Here, $1/\lambda_d$ represents the mean daily dispersal distance, conditional upon movement, and $n$ represents the number of populations in the landscape. For a given origin, $i$, the dispersal kernel entries, $m(i,)$, sum to 1. Computing $m_{ij}$ for all combinations of origins and destinations produces the movement matrix, $M$.

We also consider a zero-inflated exponential kernel (Fig 1B) which includes an additional parameter, $p_0$, to represent the daily probability that a mosquito remains in the same population. For this kernel, we have:

$$m_{ij} = \begin{cases} p_0 & , j = i \\ (1 - p_0)\frac{\exp(-\lambda_c d_{ij})}{(\sum_{j=1}^{n} \exp(-\lambda_c d_{ij})) - 1} & , j \neq i \end{cases}. \tag{2}$$

Here, $1/\lambda_c$ represents the mean daily dispersal distance, conditional upon movement, and the diagonal elements of the movement matrix, $m_{ii}$, equal $p_0$.

Default life history, demographic and dispersal parameters for *Ae. aegypti* are listed in Table 1. Given the difficulty of measuring juvenile stage mortality rates in the wild, these are chosen for consistency with observed population growth rates in the absence of density-dependence (see Sharma et al. [18] for formulae and derivations). Larval mortality increases with larval density and, according to the lumped age-class model, reaches a set value when the population is at equilibrium. Although mosquito populations vary seasonally, we assume a constant adult population size, $N_A$, for this CKMR analysis, and restrict sampling to a period of three months, corresponding to a season. Minor population size fluctuations occur in the simulation model due to sampling and stochasticity.

For dispersal, we consider movement within a 19-by-19 grid of households, based on a suburban setting such as Cairns in Queensland, Australia [25]. Mosquito traps are distributed in select households according to a specified sampling scheme (Fig 1C). In some simulations and analyses, a barrier to movement is included (Fig 1D), which could

**Table 1**. Demographic, life history and dispersal parameters for *Aedes aegypti* mosquitoes.

| Parameter: | Definition: | Value: | References: |
|---|---|---|---|
| $N_A$ | Adult population size | 25/household | [26] |
| $\mu_A$ | Adult mortality rate | 0.09/day | [27] |
| $\beta$ | Female fecundity | 20/day | [28] |
| $T_E$ | Duration of egg stage | 2 days | [29] |
| $T_L$ | Duration of larval stage | 5 days | [29] |
| $T_P$ | Duration of pupal stage | 1 day | [29] |
| $\mu_E$ | Egg mortality rate | 0.175/day | [18,30] |
| $\mu_L$ | Larval mortality rate | 0.554/day | [18] |
| $\mu_P$ | Pupal mortality rate | 0.175/day | [18,30] |
| $1/\lambda_d$ | Mean daily dispersal | 15.3 m | [11], Sect 1 in S1 Text |
| $p_0$ | Daily staying probability | [0.1, 0.9] | [25,31] |
| $1/\lambda_c$ | Mean daily dispersal conditional upon movement | [16.1, 48.4] m | [11], Sect 1 in S1 Text |
| $\delta$ | Barrier strength | [0.1, 0.9] | [25] |

represent a road, freeway or open park space. In these cases, prior to normalization, the movement rate between populations $i$ and $j$ is reduced by a factor equal to the barrier strength, $\delta$, for populations on opposite sides of the barrier, and is unchanged for populations on the same side of the barrier. Movement rates are then normalized again so that they sum to 1 for each origin, $i$.

## 2.2 Kinship probabilities

Following the CKMR methodology of Bravington et al. [9] and its application to mosquito populations by Sharma et al. [18], we now derive spatial kinship probabilities for mother-offspring and full-sibling pairs based on the lumped age-class mosquito life history model. Each kinship probability is calculated as the reproductive output consistent with that relationship divided by the total reproductive output of all adult females in that population. In each case, we consider two individuals (adult or larva) sampled at known locations, $x_1$ and $x_2$, and times, $t_1$ and $t_2$, with probability symbols and references to equations listed in Table 2. Note that mosquito sampling is lethal. Furthermore, mosquito dispersal is restricted to the adult life stage, and hence displacement between close-kin pairs represents an accumulation of adult movements between events such as emergence, egg-laying and capture, etc.

**2.2.1 Mother-offspring.** Let us begin with the simplest possible kinship probability, $P_{MOL}(x_2, t_2 | x_1, t_1)$, which represents the probability that, given an adult female sampled at location $x_1$ on day $t_1$, a larva sampled at location $x_2$ on day $t_2$ is her offspring. This can be expressed as the relative larval reproductive output at location $x_2$ on day $t_2$ of an adult female sampled at location $x_1$ on day $t_1$:

$$P_{MOL}(x_2, t_2 | x_1, t_1) = \frac{\mathbb{E}\left[\text{Larval offspring at } (x_2, t_2) \text{ from an adult female sampled at } (x_1, t_1)\right]}{\mathbb{E}\left[\text{Larval offspring at } (x_2, t_2) \text{ from all adult females}\right]} = \frac{E_{MOL}(x_2, t_2 | x_1, t_1)}{E_L(x_2)}. \tag{3}$$

Here, $E_{MOL}(x_2, t_2 | x_1, t_1)$ represents the expected number of surviving larval offspring at location $x_2$ on day $t_2$ from an adult female sampled at location $x_1$ on day $t_1$, and $E_L(x_2)$ represents the expected number of surviving larval offspring at location $x_2$ from all adult females in the population at times consistent with the time of larval sampling. Note that, since we are assuming a constant population size, $E_L(x_2)$ is independent of time and is given by:

$$E_L(x_2) = \sum_{y_2 = 0 - T_E - (T_L - 1)}^{0 - T_E} N_F(x_2) \times \beta \times (1 - \mu_E)^{T_E} \times (1 - \mu_L)^{(0 - y_2 - T_E)}. \tag{4}$$

Here, $N_F(x_2)$ represents the equilibrium adult female population size, and $y_2$ represents the day of egg-laying. Considering day 0 as the reference day (in place of $t_2$), the egg must have been laid between days $(0 - T_E - (T_L - 1))$ and $(0 - T_E)$. Eq 4 therefore represents the expected number of offspring laid by all adult females at location $x_2$ that survive the egg and larva stages up to the time of sampling (day 0).

**Table 2. Kinship categories, sampled life stages, sampling times, locations, and probability symbols used in spatial close-kin mark-recapture analysis.**

| Kinship category: | Sampled life stages: | Probability symbol: | Equations: |
|---|---|---|---|
| Mother-offspring | Adult female $(x_1, t_1)$, larva $(x_2, t_2)$ | $P_{MOL}(x_2, t_2 | x_1, t_1)$ | Sect 2.2.1 |
| | Adult female $(x_1, t_1)$, adult $(x_2, t_2)$ | $P_{MOA}(x_2, t_2 | x_1, t_1)$ | Sect 2.1 in S1 Text |
| Father-offspring | Adult male $(x_1, t_1)$, larva $(x_2, t_2)$ | $P_{FOL}(x_2, t_2 | x_1, t_1)$ | Sect 2.2 in S1 Text |
| | Adult male $(x_1, t_1)$, adult $(x_2, t_2)$ | $P_{FOA}(x_2, t_2 | x_1, t_1)$ | Sect 2.2 in S1 Text |
| Full-siblings | Larva $(x_1, t_1)$, larva $(x_2, t_2)$ | $P_{FSLL}(x_2, t_2 | x_1, t_1)$ | Sect 2.3 in S1 Text |
| | Adult $(x_1, t_1)$, adult $(x_2, t_2)$ | $P_{FSAA}(x_2, t_2 | x_1, t_1)$ | Sect 2.2.2 |
| | Larva $(x_1, t_1)$, adult $(x_2, t_2)$ | $P_{FSLA}(x_2, t_2 | x_1, t_1)$ | Sect 2.3 in S1 Text |
| | Adult $(x_1, t_1)$, larva $(x_2, t_2)$ | $P_{FSAL}(x_2, t_2 | x_1, t_1)$ | Sect 2.3 in S1 Text |

$E_{MOL}(x_2, t_2 | x_1, t_1)$, on the other hand, is specific to the sampled adult female, the location, $x_2$, and the day of larval sampling, $t_2$. This is given by:

$$E_{MOL}(x_2, t_2 | x_1, t_1) = \sum_{y_2 = t_2 - T_E - (T_L - 1)}^{t_2 - T_E} (1 - \mu_A)^{(t_1 - y_2)} \times \left( \begin{array}{c} \mathbb{I}[(t_1 - T_A) < y_2 \leq t_1] \times \psi_{MOL}(x_2, t_2 | x_1, t_1, y_2) \times \beta \\ \times (1 - \mu_E)^{T_E} \times (1 - \mu_L)^{(t_2 - y_2 - T_E)} \end{array} \right). \tag{5}$$

Here, the day of egg-laying, $y_2$, is summed over days $(t_2 - T_E - (T_L - 1))$ through $(t_2 - T_E)$, for consistency with the larva being present on the day of sampling (Fig 2). The first term in the summation represents the probability that the adult female sampled on day $t_1$ is alive on the day of egg-laying ($y_2$), and the second term (in large brackets) represents the expected surviving larval output of this adult female at location $x_2$ on day $t_2$. This latter term is equal to the probability that the larval offspring is sampled at location $x_2$ on day $t_2$ given the mother is sampled at location $x_1$ on day $t_1$ and the egg is laid on day $y_2$, $\psi_{MOL}(x_2, t_2 | x_1, t_1, y_2)$, multiplied by their daily egg production, $\beta$, multiplied by the proportion of eggs that survive the egg and larva stages from the day they were laid up to the day of sampling. An indicator function is included to limit consideration to cases where the day of egg-laying lies within the adult female's possible lifetime - i.e., between days $t_1$ and $(t_1 - T_A)$, where $T_A$ represents the maximum possible age of an adult mosquito. Although adult lifetime is exponentially-distributed, a value of $T_A$ may be chosen that captures most of this distribution and leads to accurate parameter inference.

Calculating $\psi_{MOL}(x_2, t_2 | x_1, t_1, y_2)$ requires considering a single movement type - the movement of the mother between egg-laying and sampling. Given the mother is sampled at location $x_1$ at time $t_1$, the probability that her previous location at the time of egg-laying, $y_2$, is $x_2$ is calculated by normalizing over all possible egg-laying locations, i.e.:

$$\psi_{MOL}(x_2, t_2 | x_1, t_1, y_2) = \frac{\rho(x_1, t_1 | x_2, y_2)}{\sum_{x_i} \rho(x_1, t_1 | x_i, y_2)}. \tag{6}$$

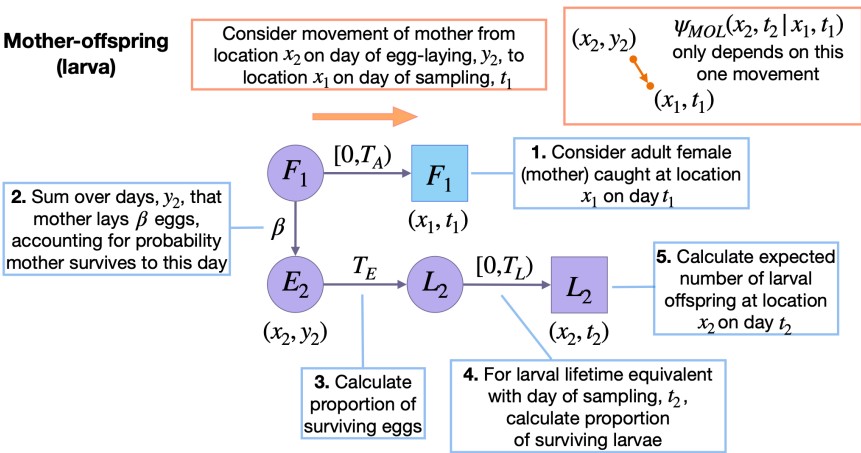

**Fig 2. Schematic representation of spatial mother-larval offspring kinship probability.** Parameters and state variables are as defined in Table 1 and Sect 2.1. Subscript 1 refers to the parent, and subscript 2 refers to the offspring (the perspective from which probabilities are calculated). Circles represent living individuals, squares represent sampled individuals, and colors represent their locations: blue for the sampled parent, $x_1$, and purple for the sampled offspring, $x_2$. Parents are sampled on day $t_1$, eggs are laid on day $y_2$, and offspring are sampled on day $t_2$. Offspring kinship probabilities are the ratio of the expected number of surviving offspring from a given adult at location $x_2$ on day $t_2$, and the expected number of surviving offspring from all adult females for this location and day. Calculating the expected number of surviving larval offspring at location $x_2$ on day $t_2$ from an adult female requires considering days of egg-laying, $y_2$, consistent with maternal ages at sampling in the range $[0, T_A)$, and larval offspring ages at sampling in the range $[0, T_L)$. The only movement to consider is that of the mother (orange arrow).

In general, $\rho(x_j, t_j | x_i, t_i)$ represents the probability that an adult mosquito is at location $x_j$ on day $t_j$, given its location is $x_i$ on day $t_i$. This is given by:

$$\rho(x_j, t_j | x_i, t_i) = (M^{(1+t_j - t_i)})_{x_i x_j}. \tag{7}$$

This is the $(x_i, x_j)$ entry ($x_i$th row and $x_j$th column) of the daily movement matrix, $M$, raised to the power, $(1+t_j - t_i)$. Recall that the entries of the movement matrix, $M$, are $m_{ij}$, as defined in Eqs 1 and 2 for the exponential and zero-inflated exponential dispersal kernel, respectively. The power accounts for the possibility that the adult female may move on each day between egg-laying (or in later calculations, emergence) and sampling, inclusive of these two days.

Extending the mother-larval offspring kinship probability to the mother-adult offspring case is described in Sect 2.1 in S1 Text. Extensions to father-offspring cases are described in Sect 2.2 in S1 Text.

**2.2.2 Full-siblings.** Next, we consider the full-sibling kinship probability for adult-adult pairs, $P_{FSAA}(x_2, t_2 | x_1, t_1)$, which represents the probability that, given an adult sampled at location $x_1$ on day $t_1$, an adult sampled at location $x_2$ on day $t_2$ is their full-sibling. This can be expressed as the relative adult reproductive output at location $x_2$ on day $t_2$ of the mother of a larva sampled at location $x_1$ on day $t_1$:

$$P_{FSAA}(x_2, t_2 | x_1, t_1) = \frac{\mathbb{E}[\text{Adults at } (x_2, t_2) \text{ that are full-siblings of an adult sampled at } (x_1, t_1)]}{\mathbb{E}[\text{Adult offspring at } (x_2, t_2) \text{ from all adult females}]} = \frac{E_{FSAA}(x_2, t_2 | x_1, t_1)}{E_A(x_2)}. \tag{8}$$

Here, $E_{FSAA}(x_2, t_2 | x_1, t_1)$ represents the expected number of surviving adults at location $x_2$ on day $t_2$ that are full-siblings of an adult sampled at location $x_1$ on day $t_1$, and $E_A(x_2)$ represents the expected number of surviving adult offspring at location $x_2$ from all adult females at times consistent with the time of adult offspring sampling. Assuming a population at equilibrium, $E_A(x_2)$ is independent of time and is given by:

$$E_A(x_2) = \sum_{y_2 = 0 - T_E - T_L - T_P - (T_A - 1)}^{0 - T_E - T_L - T_P} N_F(x_2) \times \beta \times (1 - \mu_E)^{T_E} \times (1 - \mu_L)^{T_L} \times (1 - \mu_P)^{T_P} \times (1 - \mu_A)^{(0 - y_2 - T_E - T_L - T_P)}. \tag{9}$$

For convenience, let us refer to the adult sampled on day $t_1$ as individual 1. To calculate $E_{FSAA}(x_2, t_2 | x_1, t_1)$, we treat the day that egg 1 is laid, $y_1$, as a latent variable and take an expectation over it:

$$E_{FSAA}(x_2, t_2 | x_1, t_1) = \sum_{y_1 = t_1 - T_E - T_L - T_P - (T_A - 1)}^{t_1 - T_E - T_L - T_P} p_A(t_1 - y_1 - T_E - T_L - T_P) \times E_{FSAA}(x_2, t_2 | x_1, t_1, y_1). \tag{10}$$

Here, the expectation over the day that egg 1 is laid, $y_1$, is taken over days $(t_1 - T_E - T_L - T_P - (T_A - 1))$ through $(t_1 - T_E - T_L - T_P)$, for consistency with the day that larva 1 is sampled (Fig 3). The term $E_{FSAA}(x_2, t_2 | x_1, t_1, y_1)$ represents the expected number of surviving adults at location $x_2$ on day $t_2$ that are full-siblings of adult 1, conditional upon egg 1 being laid on day $y_1$. Additionally, $p_A(t_1 - y_1 - T_E - T_L - T_P)$ represents the probability that egg 1 is laid on day $(t_1 - y_1 - T_E - T_L - T_P)$. In general, $p_A(t)$ represents the probability that a given adult in the population has age $t$ which, following from the daily adult survival probability, $(1 - \mu_A)$, is given by:

$$p_A(t) = (1 - \mu_A)^t \Big/ \sum_{t_j = 0}^{T_A - 1} (1 - \mu_A)^{t_j}. \tag{11}$$

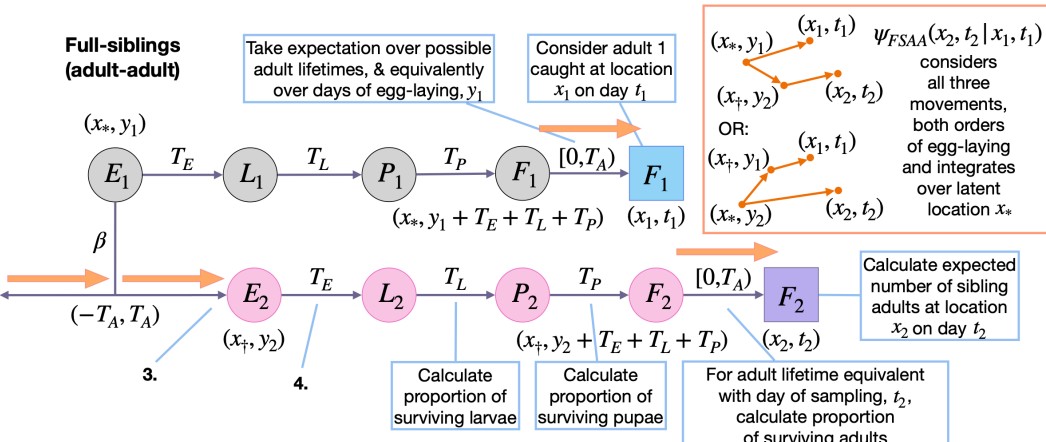

**Fig 3. Schematic representation of spatial adult-adult full-sibling kinship probabilities.** Parameters and state variables are as defined in Table 1 and Sect 2.2. Subscript 1 refers to the reference sibling, and subscript 2 refers to the sibling from whose perspective the probabilities are calculated. Circles represent living individuals, squares represent sampled individuals, and colors represent their locations: blue for sibling 1, $x_1$, purple for sibling 2, $x_2$, grey for the location of egg-laying for sibling 1, and pink for the location of egg-laying for sibling 2. The location of egg-laying for the firstborn sibling (here, sibling 1) is denoted by $x_*$, and $x_†$ denotes the egg-laying location for the other sibling. Sibling 1 is sampled on day $t_1$ and laid on day $y_1$. Sibling 2 is sampled on day $t_2$ and laid on day $y_2$. Sibling kinship probabilities are the ratio of the expected number of surviving siblings of a given individual at location $x_2$ on day $t_2$, and the expected number of surviving offspring from all adult females for this location and day. Calculating the expected number of surviving full-siblings at location $x_2$ on day $t_2$ requires considering days of egg-laying, $y_1$ and $y_2$, consistent with adult ages at sampling in the range $[0, T_A)$. There are three movements to consider: those of the mother and two adult siblings (orange arrows). Movement probabilities consider both orders of egg-laying.

The term $E_{FSAA}(x_2, t_2|x_1, t_1, y_1)$ represents the expected number of surviving adults at location $x_2$ on day $t_2$ that are full-siblings of adult 1, conditional upon egg 1 being laid on day $y_1$. This is given by:

$$E_{FSAA}(x_2, t_2|x_1, t_1, y_1) = \frac{1}{2} \sum_{y_2=y_1-(T_A-1)}^{y_1+(T_A-1)} (1-\mu_A)^{|y_2-y_1|}$$

$$\times \left( \begin{array}{c} \mathbb{I}[(t_2 - T_E - T_L - T_P - T_A) < y_2 \le (t_2 - T_E - T_L - T_P)] \\ \times \psi_{FSAA}(x_2, t_2|x_1, t_1, y_1, y_2) \times \beta \times (1-\mu_E)^{T_E} \\ \times (1-\mu_L)^{T_L} \times (1-\mu_P)^{T_P} \times (1-\mu_A)^{(t_2-y_2-T_E-T_L-T_P)} \end{array} \right). \tag{12}$$

Here, the day of sibling egg-laying, $y_2$, is summed over days $(y_1 - (T_A - 1))$ through $(y_1 + (T_A - 1))$, for consistency with the mother's potential lifespan (Fig 3), and considering that either of the larval offspring may have been laid first. Since we are summing two equally-weighted scenarios regarding offspring order, we include a multiplier of 1/2 in the expectation. The first term within the summation then represents the probability that the mother is alive on the day of sibling egg-laying, with the absolute value, $|y_2 - y_1|$, accounting for both offspring orders. The second term (in larger brackets) represents the expected adult offspring output of the mother of adult 1 at location $x_2$ on day $t_2$. This is the same equation as for the mother-larval offspring case with three exceptions: i) daily egg production is multiplied by the proportion of eggs that survive the egg, larva, pupa and adult stages up to the day of sampling to reflect the fact that adults rather than larvae are being sampled, ii) the indicator function limits consideration to cases where the day of sibling egg-laying, $y_2$, is between days $(t_2 - T_E - T_L - T_P - (T_A - 1))$ and $(t_2 - T_E - T_L - T_P)$, for consistency with an adult sibling being sampled on day $t_2$, and iii) there are now three movements captured by the composite movement term, $\psi_{FSAA}(x_2, t_2|x_1, t_1, y_1, y_2)$, which represents

the probability that adult offspring 2 is sampled at location $x_2$ on day $t_2$ given that adult offspring 1 is sampled at location $x_1$ on day $t_1$, egg 1 is laid on day $y_1$, and egg 2 is laid on day $y_2$.

Calculating $\psi_{FSAA}(x_2, t_2|x_1, t_1, y_1, y_2)$ requires considering the mother's movement between egg-laying events, in addition to the movement of both adult offspring for both offspring orders, i.e.:

$$\psi_{FSAA}(x_2, t_2|x_1, t_1, y_1, y_2) = \sum_{x_*} \left( \frac{\rho(x_1, t_1|x_*, (\min\{y_1, y_2\} + T_E + T_L + T_P))}{\sum_{x_i} \rho(x_1, t_1|x_i, (\min\{y_1, y_2\} + T_E + T_L + T_P))} \right)$$
$$\times \rho(x_2, t_2|x_*, (\min\{y_1, y_2\} + T_E + T_L + T_P)). \tag{13}$$

Here, we take an expectation over a latent egg-laying location for the firstborn sibling, $x_*$, and multiply the probability that the firstborn sibling is laid at location $x_*$ given that adult sibling 1 is sampled at location $x_1$ by the probability that adult sibling 2 is sampled at location $x_2$ (the former probability requires normalizing over all egg-laying locations). For both adult siblings, movement begins after development through the egg, larva and pupa life stages (i.e., after $(T_E + T_L + T_P)$ days), and the mother's movement between egg-laying events is incorporated by effectively adding $|y_2 - y_1|$ days of movement either backwards or forwards in time, depending on the offspring order.

Extending full-sibling kinship probabilities to other life stage pairs is relatively straightforward. We consider the case of larva-larva, larva-adult and adult-larva full-sibling pairs in Sect 2.3 in S1 Text.

## 2.3 Pseudo-likelihood calculation

The goal of this spatial CKMR analysis is to make inferences about dispersal parameters given data on the frequency, timing and location of observed close-kin pairs. Here, we calculate the likelihood of parent-offspring and full-sibling pairs in a manner that takes advantage of the nature of the kinship probabilities and the sampling process. The kinship probabilities for each pair of individuals are assumed to be independent of each other, even though they are not. For this reason the combined likelihood is referred to as a "pseudo-likelihood" [9] - an approach that has been shown to produce accurate parameter and variance estimates provided the size of each sampling event is sufficiently low relative to the total population size [32,33].

**2.3.1 Parent-offspring pairs.** Let us begin by considering the mother-larval offspring kinship probability, $p_{MOL}(x_2, t_2|x_1, t_1)$, which represents the probability that, given an adult female sampled at location $x_1$ on day $t_1$, a given larva sampled at location $x_2$ on day $t_2$ is her offspring. Now consider $n_F(x_1, t_1)$ adult females sampled at location $x_1$ on day $t_1$. The probability that a larva sampled at location $x_2$ on day $t_2$ has a mother amongst the $n_F(x_1, t_1)$ sampled adult females, $p_{MOL}(x_2, t_2|x_1, t_1)$, is equal to one minus the probability that none of the $n_F(x_1, t_1)$ sampled adult females are the larva's mother, i.e.:

$$p_{MOL}(x_2, t_2|x_1, t_1) = 1 - (1 - P_{MOL}(x_2, t_2|x_1, t_1))^{n_F(x_1, t_1)}. \tag{14}$$

Here, $P_{MOL}(x_2, t_2|x_1, t_1)$ is as defined in Eq 3. Now consider $n_L(x_2, t_2)$ larvae sampled at location $x_2$ on day $t_2$, and let $k_{MOL}(x_2, t_2|x_1, t_1)$ be the number of larvae sampled at location $x_2$ on day $t_2$ that have a mother amongst the adult females sampled at location $x_1$ on day $t_1$. The pseudo-likelihood that $k_{MOL}(x_2, t_2|x_1, t_1)$ of the $n_L(x_2, t_2)$ larvae sampled at location $x_2$ on day $t_2$ have a mother amongst the adult females sampled at location $x_1$ on day $t_1$ follows from the binomial distribution:

$$L(k_{MOL}(x_2, t_2|x_1, t_1)) = \binom{n_L(x_2, t_2)}{k_{MOL}(x_2, t_2|x_1, t_1)} \times \left( \begin{array}{c} p_{MOL}(x_2, t_2|x_1, t_1)^{k_{MOL}(x_2, t_2|x_1, t_1)} \times \\ (1 - p_{MOL}(x_2, t_2|x_1, t_1))^{n_L(x_2, t_2) - k_{MOL}(x_2, t_2|x_1, t_1)} \end{array} \right). \tag{15}$$

The full log-pseudo-likelihood for mother-larval offspring pairs, $\Lambda_{MOL}$, follows from summing the log-pseudo-likelihood over all adult female sampling days, $t_1$, over consistent larval offspring sampling days, $t_2$, and over all adult female and

larval sampling locations, $x_1$ and $x_2$, respectively:

$$\Lambda_{MOL} = \sum_{x_1} \sum_{t_1} \sum_{x_2} \sum_{t_2=t_1+T_E+T_L+T_P-T_A}^{t_1+T_E+T_L+T_P+T_A} \left( \begin{array}{l} k_{MOL}(x_2,t_2|x_1,t_1) \log p_{MOL}(x_2,t_2|x_1,t_1) + \\ (n_L(x_2,t_2) - k_{MOL}(x_2,t_2|x_1,t_1)) \log (1 - p_{MOL}(x_2,t_2|x_1,t_1)) \end{array} \right). \tag{16}$$

Note that, for the purpose of parameter interference, we can drop the first term in the pseudo-likelihood equation, and for the purpose of efficient computation, we consider consistent adult sampling days from $(t_1 + T_E + T_L + T_P - (T_A - 1))$ through $(t_1 + T_E + T_L + T_P + (T_A - 1))$. The earliest adult sampling day (relative to $t_1$) corresponds to the case where the mother laid the offspring at the beginning of her life, was sampled at the end of her life, and the adult offspring was sampled at the beginning of its life. The latest adult sampling day (relative to $t_1$) corresponds to the case where the mother was sampled on the day they laid their offspring, and the adult offspring was sampled at the end of its life.

Parent-offspring pseudo-likelihood equations for other sampled sexes and life stages follow an equivalent formulation. The main point to note is that consistent offspring sampling days are specific to the kinship and sampled life stages being considered (these can be deduced from event history diagrams like those in Fig 2). For adult offspring cases where $x_1 = x_2$ and $t_1 = t_2$, the number of sampled adults, $n_A(x_2, t_2)$, is reduced by one to account for the fact that an adult cannot be its own parent. The joint log-pseudo-likelihood for parent-offspring pairs is then given by:

$$\Lambda_{PO} = \Lambda_{MOL} + \Lambda_{MOA} + \Lambda_{FOL} + \Lambda_{FOA}. \tag{17}$$

Here, $\Lambda_{MOA}$, $\Lambda_{FOL}$ and $\Lambda_{FOA}$ denote the log-pseudo-likelihoods for mother-adult offspring pairs, father-larval offspring pairs and father-adult offspring pairs, respectively.

**2.3.2 Full-sibling pairs.** For full-siblings, we begin with the adult-adult full-sibling kinship probability, $P_{FSAA}(x_2, t_2|x_1, t_1)$, defined in Eq 8, which represents the probability that, given an adult sampled at location $x_1$ on day $t_1$, an adult sampled at location $x_2$ on day $t_2$ is their full-sibling. We consider a given adult, indexed by $i$ and sampled at location $x_1(i)$ on day $t_1(i)$, and $n_A(x_2, t_2)$ adults sampled at location $x_2$ on day $t_2$. Let $k_{FSAA}(i, x_2, t_2)$ be the number of adults sampled at location $x_2$ on day $t_2$ that are full-siblings of adult $i$. The pseudo-likelihood that $k_{FSAA}(i, x_2, t_2)$ of the $n_A(x_2, t_2)$ sampled adults at location $x_2$ on day $t_2$ are full-siblings of adult $i$ follows from the binomial distribution:

$$L(k_{FSAA}(i,x_2,t_2)) = \binom{n_A(x_2,t_2)}{k_{FSAA}(i,x_2,t_2)} \times P_{FSAA}(x_2,t_2|x_1(i),t_1(i))^{k_{FSAA}(i,x_2,t_2)} \\ \times (1 - P_{FSAA}(x_2,t_2|x_1(i),t_1(i)))^{n_A(x_2,t_2)-k_{FSAA}(i,x_2,t_2)}. \tag{18}$$

Note that, for cases where $x_1(i) = x_2$ and $t_1(i) = t_2$, the number of sampled adults at location $x_2$ on day $t_2$, $n_A(x_2, t_2)$, is reduced by one to account for the fact that an adult cannot be its own sibling. Additionally, when counting siblings, we only consider siblings with indices $>i$ to avoid double-counting and self-counting. The full log-pseudo-likelihood for adult-adult full-sibling pairs, $\Lambda_{FSAA}$, follows from summing the log-pseudo-likelihood over all sampled adults, $i$, over all sampled locations, $x_2$, and over consistent adult sampling days, $t_2$:

$$\Lambda_{FSAA} = \sum_{i=1}^{n_A-1} \sum_{x_2} \sum_{t_2=t_1(i)-2(T_A-1)}^{t_1(i)+2(T_A-1)} k_{FSAA}(i,x_2,t_2) \log P_{FSAA}(x_2,t_2|x_1(i),t_1(i)) \\ + (n_A(x_2,t_2) - k_{FSAA}(i,x_2,t_2)) \log (1 - P_{FSAA}(x_2,t_2|x_1(i),t_1(i))). \tag{19}$$

Full-sibling pairs represent the majority of the computational burden of the pseudo-likelihood calculation, and from the above equation, this can be seen to scale approximately linearly with total sample size. Consistent adult sampling days for the full-sibling case are from $(t_1(i)-2(T_A-1))$ through $(t_1(i)+2(T_A-1))$. The earliest adult sampling day (relative to $t_1(i)$) corresponds to the case where the mother laid individual 2 at the beginning of her life and individual 1 at the end of her life,

adult 1 was sampled at the end of its life, and adult 2 was sampled soon after emergence. The latest adult sampling day (relative to $t_1(i)$) corresponds to the reverse case. Full-sibling pseudo-likelihood equations for other life stage pairs follow an equivalent formulation, with consistent sampling days specific to the kinship and sampled life stages being considered (these can be deduced from event history diagrams like those in Fig 3). The joint log-pseudo-likelihood for full-sibling pairs is then given by:

$$\Lambda_{FS} = \Lambda_{FSLL} + \Lambda_{FSLA} + \Lambda_{FSAL} + \Lambda_{FSAA}. \tag{20}$$

Here, $\Lambda_{FSLL}$, $\Lambda_{FSLA}$ and $\Lambda_{FSAL}$ denote the log-pseudo-likelihoods for larva-larva, larva-adult and adult-larva full-sibling pairs, respectively.

**2.3.3 Parameter inference.** Despite parent-offspring and full-sibling kinship probabilities not being independent, the pseudo-likelihood approach enables us to combine these likelihoods, provided the size of each sampling event is sufficiently low relative to the total population size [9]. As we will see later, our simulation studies suggest this to be the case. We therefore combine these log-pseudo-likelihoods to obtain a log-pseudo-likelihood for the entire data set:

$$\Lambda = \Lambda_{PO} + \Lambda_{FS}. \tag{21}$$

Parameter inference can then proceed by varying a subset of the dispersal and/or demographic parameters in Table 1 in order to minimize $-\Lambda$. We used the **nlminb** function implemented in the **optimx** function in R [34] to perform our optimizations. Parameter identifiability was not an apparent issue in our analyses; however, Bravington et al. [9] note that the Fisher information matrix can be used to study parameter identifiability in CKMR analyses, and that the per-pair Fisher information can provide a valuable tool for sampling scheme design.

## 2.4 Individual-based simulation model

We used a previously-developed simulation package, **mPlex** [18], to model mosquito life history and to test the effectiveness of the CKMR approach at estimating mosquito dispersal and demographic parameters. The model is an individual-based adaptation of a previous model, **MGDrivE** [35], which is a genetic and spatial extension of the lumped age-class model applied to mosquitoes by Hancock and Godfray [21] and Deredec et al. [22] (Fig 1A). Our previous application of **mPlex** to CKMR-based inference problems considered only panmictic populations; however, the functionality had already been included to account for spatial population structure, with mosquitoes being distributed across populations in a metapopulation [35]. Each population has an equilibrium adult population size, $N_A^*$, and exchanges migrants with the other populations. Populations are partitioned according to discrete life stages - egg, larva, pupa and adult - with sub-adult stages having fixed durations as defined earlier. See Sharma et al. [18] for more details.

## 3 Results

We used simulated data from the individual-based mosquito model to explore the feasibility of spatial CKMR to infer dispersal parameters for *Ae. aegypti*. Our simulated metapopulation consisted of a 19-by-19 grid of households (Fig 1C) each inhabited by 25 adult mosquitoes at equilibrium with bionomic parameters listed in Table 1. Landscape dimensions were chosen to accommodate the trap arrangements described below, as well as a buffer width of at least three non-trap nodes along each landscape edge (e.g., Fig 1D) to reduce boundary effects. Open questions concern the optimal sampling scheme to estimate dispersal parameters for *Ae. aegypti* using spatial CKMR methods, and the range of dispersal parameters that can be accurately estimated. To address these questions, we first explored logistically-feasible sampling schemes to estimate mean dispersal distance by varying: i) sampled life stage (larva or adult), ii) total sample size (1,000-3,000 sequenced individuals), and iii) the number and spacing of trap nodes (arranged in 4-by-4, 5-by-5 or 6-by-6 grids with zero, one or two population nodes separating each trap node). Based on a previous analysis to estimate

mosquito demographic parameters using CKMR [18], our adult samples were of females (since mosquito traps are often tailored to this sex), sampling frequency was biweekly (i.e., twice per week, as is often the case for mosquito surveillance programs [36]), and sampling duration was for three months (corresponding to a season). Our likelihood calculations were based on parent-offspring and full-sibling pairs. Half-siblings and higher-order kinship pairs could be included for individual data analyses; however, computational burden prevented us from including these for exploratory analyses (note that only paternal half-siblings are possible, since we assume that females mate only once). Initially, we focused on estimating mean daily dispersal distance, $1/\lambda_d$, and for subsequent analyses, also estimated daily staying probability, $p_0$, and barrier strength, $\delta$.

### 3.1 Optimal sampling scheme to estimate daily dispersal distance

To estimate mean daily dispersal distance, our default sampling scheme consisted of a total of 1,000 sequenced individuals sampled biweekly over a three-month period spread over a 6-by-6 grid of trap nodes with one population node separating each trap node (Fig 1C) (i.e., ca. 1 individual sampled twice per week from each trap node, for a total of 1,000 individuals across all trap nodes after three months of sampling). We first explored the optimal distribution of sampled life stage to estimate $1/\lambda_d$, exploring three scenarios: all larvae, all adult females, and half larvae/half adult females. Results of 100 simulation-and-analysis replicates for each scenario are depicted in Sect 3.1 in S1 Text. These suggest that all three life stage scenarios result in adequate parameter inference, in terms of both accuracy of the median and tightness of the interquartile range (IQR), albeit consistently underestimating $1/\lambda_d$ by ca. 10-15%. Since most mosquito traps are designed to target adult females, we proceeded with modeling samples of this sex and life stage. While larval samples also produce adequate inference, either exclusively or together with adult female samples, larvae are more difficult to sample in the environment due to breeding sites sometimes being hidden or inaccessible. The problem of larval sampling is exacerbated for CKMR studies due to the requirement that individuals be sampled independently. Therefore, if multiple larvae are collected in a larval dip from a single breeding site, only one can be used in the analysis to prevent biasing the number of collected sibling pairs upwards, thus further increasing the effort required to achieve a substantial larval sample.

Next, we explored the optimal sample size to estimate $1/\lambda_d$ for *Ae. aegypti*. We performed 100 simulation-and-analysis replicates for each of five total sample sizes - 1,000, 1,500, 2,000, 2,500 and 3,000 adult females - depicted in Sect 3.1 in S1 Text. Results suggest that estimates of $1/\lambda_d$ become more precise for larger sample sizes (as measured by the IQR); but there are diminishing returns in precision for sample sizes larger than 2,000. The median estimate of $1/\lambda_d$ remains underestimated by ca. 10-15% regardless of sample size. We therefore proceed with an optimal sample size of 2,000 adult females, collected biweekly over a three-month period. This equates to ca. 2 individuals sampled twice per week from each trap node over the three-month period.

Next, we explored the optimal number and spacing of trap nodes to estimate $1/\lambda_d$ for *Ae. aegypti*. We performed 100 simulation-and-analysis replicates for trap nodes arranged in 4-by-4, 5-by-5 or 6-by-6 grids with zero, one or two population nodes separating each trap node (except for the case of a 6-by-6 grid of trap nodes where the simulated landscape could only accommodate zero or one population node separating each trap node). Results, depicted in Fig 4A, suggest that when traps are optimally spread out throughout a landscape, inference of mean dispersal distance is very accurate (as measured by the difference between the median and true value). Notably, estimates of $1/\lambda_d$ are unbiased for trap nodes arranged in a 5-by-5 grid with two population nodes separating each trap node, and are relatively close to the true value for a smaller number of trap nodes (e.g., arranged in a 4-by-4 grid with two population nodes separating each trap node), or for a larger number of trap nodes that are closer together (e.g., arranged in a 6-by-6 grid with one population node separating each trap node). Conversely, estimates of $1/\lambda_d$ are highly biased when traps are clustered together, as can be seen for grids of 4-by-4, 5-by-5 or 6-by-6 adjacent trap nodes. In these cases, the median estimate of $1/\lambda_d$ is ca. 30% less than the true value. This is an intuitive result, as when trap nodes are clustered together, they are more likely

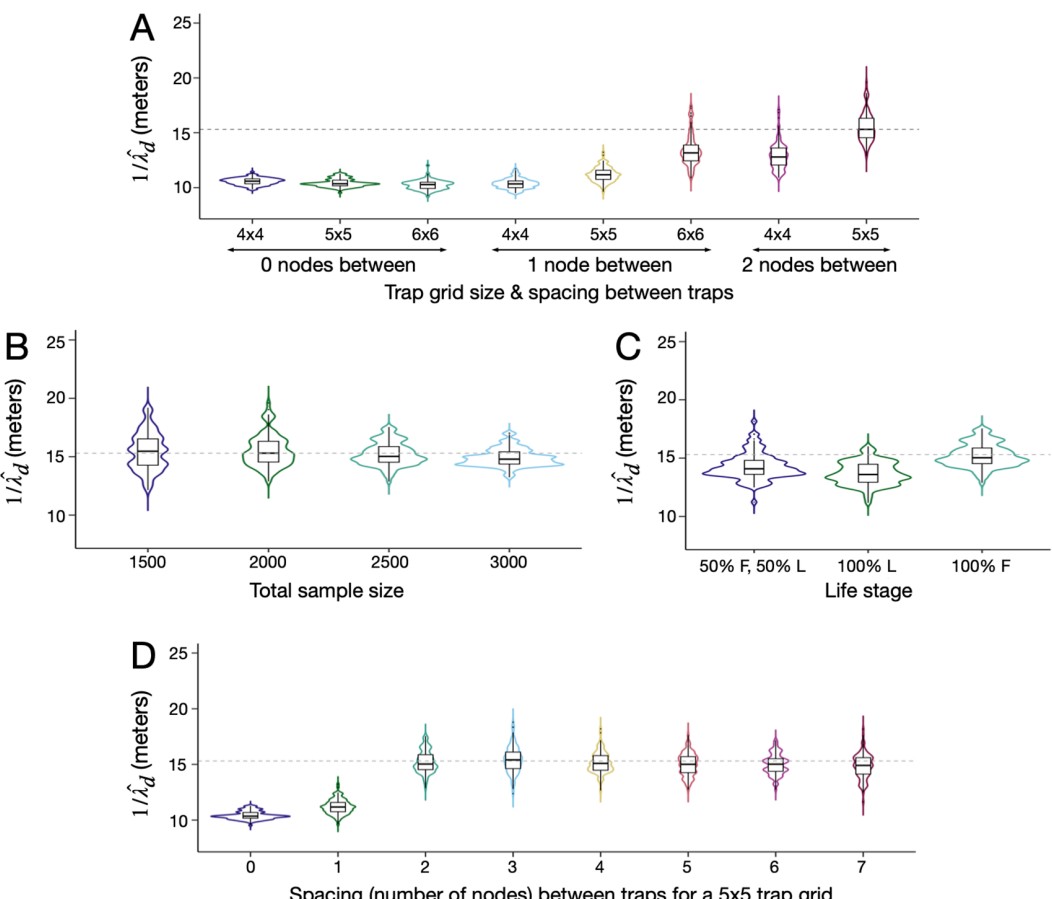

**Fig 4**. **Sampling schemes to estimate** $1/\lambda_d$ **for *Ae. aegypti*.** Violin plots depict estimates of $1/\lambda_d$ for sampling scenarios described in Sect 3.1. The default simulated metapopulation consists of a 19-by-19 grid of households each inhabited by 25 adult *Ae. aegypti* at equilibrium with bionomic parameters listed in Table 1. Boxes depict median and interquartile ranges of 100 simulation-and-analysis replicates for each scenario, thin lines represent 5% and 95% quantiles, points represent outliers, and kernel density plots are superimposed. The initial sampling scheme consists of a total of 2,000 adult females sampled as ca. 2 individuals collected twice weekly over a three-month period for each trap node, considering a 6-by-6 array of trap nodes with one population node separating each trap node (Fig 1C). In panel **(A)**, the number and spacing of trap nodes is varied (arranged in 4-by-4, 5-by-5 or 6-by-6 grids with zero, one or two population nodes separating each trap node). In panel **(B)**, trap nodes are arranged in a 5-by-5 grid with two population nodes separating each trap node (Fig 1D), and total sample sizes of 1,500, 2,000, 2,500 and 3,000 are explored. In panel **(C)**, a sample size of 2,500 is adopted, and three life stage proportions are explored: all larvae, all adult females, and half larvae/half adult females. The optimal sampling scheme consists of 2,500 adult females collected biweekly over a three-month period spread over a 5-by-5 grid of trap nodes with two population nodes separating each trap node. In panel **(D)**, the optimal sampling scheme is adopted and a larger metapopulation consisting of a 37-by-37 grid of households is used to evaluate how far apart trap nodes may be placed (separated by 0-7 household nodes) while still obtaining reasonable estimates of $1/\lambda_d$.

to capture nearby movement but less likely to capture distant movement, hence leading to underestimates of mean daily dispersal. The optimal sampling scenario where trap nodes are separated by two population nodes equates to a distance between trap nodes of 49.8 meters (since population nodes are separated by 16.6 meters in these simulations), which is approximately equal to the mean lifetime dispersal distance of *Ae. aegypti* mosquitoes used to parameterize this model of 45.2 meters [11]. Of note, the 5-by-5 grid of trap nodes with each trap node separated by two population nodes represents the largest number of traps that can be accommodated on the simulated landscape for this degree of separation.

Finally, given the significance of the number and spacing of trap nodes for estimating $1/\lambda_d$ evident in Fig 4A, we revisited the optimal sample size and life stage distribution given a 5-by-5 grid of trap nodes with each trap node separated by

two population nodes. Results for sample size again suggest that samples of 2,000 adult females produce adequate estimates of $1/\lambda_d$; but also suggest an improvement in precision (as measured by IQR) for a population size of 2,500 (Fig 4B). We therefore proceed with this total sample size, which equates to ca. 3-4 individuals sampled twice per week for each trap node over the three-month period. Regarding the optimal distribution of sampled life stage, results again suggest that all three life stage scenarios result in comparable parameter inference (as measured by the accuracy of the mean and the tightness of the IQR) (Fig 4C). We therefore continue with an adult female sample, as per our previous reasoning. With all of these considerations in mind, the optimal sampling scheme therefore consists of a total of 2,500 adult females collected biweekly over a three-month period spread over a 5-by-5 grid of trap nodes with two population nodes separating each trap node.

Considering the optimal sampling scheme corresponds to the case where a 5-by-5 grid of traps is spread out as much as possible through a 19-by-19 grid of households (Fig 1D), we conducted a supplemental analysis in which trap nodes were separated by 0-7 population nodes, to see how far apart traps may be placed while still obtaining reasonable estimates of mean daily dispersal distance. To achieve this, we carried out 100 simulation-and-analysis replicates for each scenario on a larger 37-by-37 household grid, in order to accommodate the maximum trap separation and provide a two-house buffer at each landscape boundary. Results, depicted in Fig 4D, suggest that estimates of $1/\lambda_d$ are reasonably accurate (i.e., the IQR encompasses the true value) for trap node separations of 2-7 household nodes, although the accuracy (as measured by the distance between the median of 100 simulations and the true value) begins to decline for trap node separations of 6-7 household nodes. This suggests that accurate estimation of $1/\lambda_d$ may be obtained for trap nodes placed a distance of 1-2.5 times the mean lifetime dispersal distance apart. Conveniently, this implies that the dispersal behavior of the species need only be approximately known when designing a sampling scheme.

**3.1.1 False negative and false positive kinship inference.** We explored the role that false negative and false positive kinship inference could have on estimation of $1/\lambda_d$. To investigate this, we used the optimal sampling scheme determined above for a 19-by-19 household grid, and performed 100 simulation replicates for scenarios in which: i) 0-20% of mother-offspring and full-sibling pairs were decoupled at random (i.e., introducing false negatives), and ii) 0-20% of individuals without sampled mothers or full-siblings were assigned them at random (i.e., introducing false positives). Results, depicted in Fig 5, show that introduction of false negatives, even at proportions as high as 20%, has little impact on the estimated value of $1/\lambda_d$ (Fig 5A). Introduction of false positives, however, causes an upwards bias in estimates of $1/\lambda_d$ by a degree proportional to the number of false positives in the dataset (Fig 5B). This is likely because false positives tend to be separated by larger distances than true positives. False negatives, on the other hand, convey little information about dispersal patterns. Based on these results, we recommend aiming for a false positive kinship inference rate of less than 5%, and erring towards false negatives rather than false positives where trade-offs in kinship inference exist.

## 3.2 CKMR-based estimates of barrier strength and daily staying probability

Given the optimal sampling scheme, we employ the flexibility of the formal CKMR approach to estimate additional parameters describing more complex dispersal patterns - the strength of a barrier to movement, $\delta$, and the daily staying probability for a zero-inflated exponential kernel, $p_0$. To begin, we consider a barrier to movement, which could represent a road, freeway or open park space for *Ae. aegypti*, as has been documented as being important for dispersal of this species [25]. We depict the barrier as a line through the landscape (Fig 1D), whereby movement to the other side of the barrier is reduced by a factor, $\delta$, and movement on the same side of the barrier is unaltered. We explore $\delta$ values in the range [0.1,0.9] and adopt the optimal sampling scheme determined in Sect 3.1 - a total sample of 2,500 adult females sampled from a 5-by-5 grid of trap nodes with two population nodes separating each trap node. Results in Fig 6C suggest that estimates of $\delta$ are very accurate (as measured by the distance between the median of 100 simulations and the true value) for $\delta \geq 0.5$, and barrier strength estimates become more precise (as measured by IQR) for $\delta \geq 0.75$. Estimates of barrier

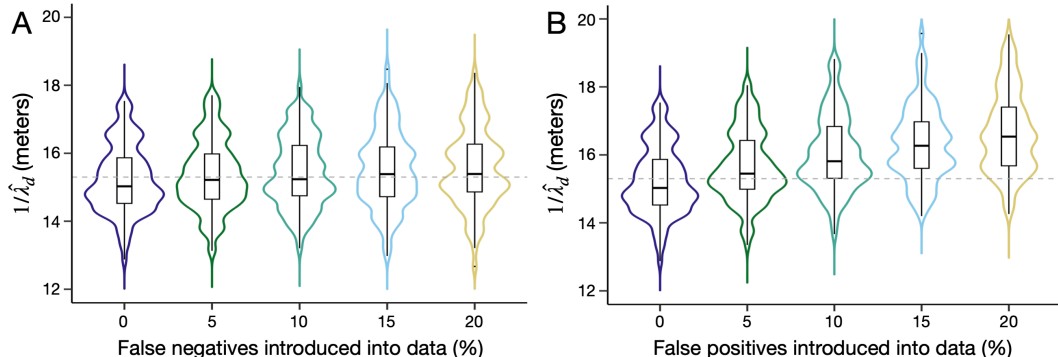

**Fig 5. Influence of false negative and false positive kinship pairs on estimation of** $1/\lambda_d$**.** Violin plots depict estimates of mean daily dispersal distance, $1/\lambda_d$, obtained using the spatial CKMR approach for the optimal sampling scheme determined in Sect 3.1. Two scenarios are explored: **(A)** in which 0-20% of mother-offspring and full-sibling pairs were decoupled at random (i.e., introducing false negatives), and **(B)** in which 0-20% of individuals without sampled mothers or full-siblings were assigned them at random (i.e., introducing false positives). The simulated metapopulation consists of a 19-by-19 grid of households each inhabited by 25 adult *Ae. aegypti* at equilibrium with bionomic parameters listed in Table 1. Boxes depict median and interquartile ranges of 100 simulation-and-analysis replicates for each scenario, thin lines represent 5% and 95% quantiles, points represent outliers, and kernel density plots are superimposed. The true value of $1/\lambda_d$ is depicted by a dotted line.

strength tend to be slightly overestimated for $\delta \leq 0.25$; but the true value still falls within the IQR. For field data where a value of $\delta \leq 0.25$ is inferred, we advise also fitting a model without a barrier present and performing model selection. Estimates of $1/\lambda_d$ are reasonably accurate when barrier strength is simultaneously estimated (Fig 6A), with the true value consistently falling within the IQR; however, the median estimate from 100 simulations falls below the true value by ca. 3-5% for barrier strengths $\geq 0.75$. This is perhaps not surprising, considering that the presence of a strong barrier in a landscape reduces the mean distance traveled by an individual over their lifetime. Fortunately, barrier location does not present an obstacle to parameter inference, as we demonstrate in Sect 3.2 in S1 Text.

Next, we consider a zero-inflated exponential dispersal kernel and estimate the daily staying probability, $p_0$, alongside the mean daily dispersal distance conditional upon movement, $\lambda_c$. Exploring this kernel is motivated by the fact that several mosquito species, such as *Ae. aegypti*, obtain most of their resources from a small area such as a household and tend to remain at this location for the majority of their lifetime [5]. The zero-inflated exponential dispersal kernel is formulated in Eq 2, and dispersal kernels having $p_0 \in [0.1, 0.9]$ are depicted in Fig 1B. For each value of $p_0$, we maintain a mean daily dispersal distance of 15.3 m (Table 1), which translates to a daily dispersal distance conditional upon movement, $1/\lambda_c$, of 16.1 m for $p_0 = 0.1$, 17.7 m for $p_0 = 0.25$, 21.6 m for $p_0 = 0.5$, 30.6 m for $p_0 = 0.75$, and 48.4 m for $p_0 = 0.9$. Again, we assume the optimal sampling scheme determined in Sect 3.1. Results in Fig 6B suggest that estimates of $1/\lambda_c$ are accurate and precise for all explored values of $p_0$, as measured by the difference between the median of 100 simulations and the true value of $1/\lambda_c$, and by the IQR. On the other hand, $p_0$ can be accurately and precisely estimated for true $p_0 \geq 0.75$; but smaller true values of $p_0$ are overestimated by increasingly large degrees (as measured by the median of 100 simulation replicates) - a true value of $p_0$ of 0.5 is overestimated by ca. 0.09, a true value of 0.25 is overestimated by ca. 0.20, and a true value of 0.1 is overestimated by ca. 0.36 (Fig 6D). Overestimates of $p_0$ may be related to the calculation of movement probabilities on a grid landscape. For the 19-by-19 landscape depicted in Fig 1C, an exponential kernel without zero-inflation has diagonal entries of its transition matrix (effectively, staying probabilities) between 0.18 and 0.38, and hence staying probabilities of a zero-inflated kernel with $p_0 \in \{0.1, 0.25\}$ may be difficult to discern. For field data where a value of $p_0 \leq 0.5$ is inferred, we advise also fitting an exponential dispersal kernel and performing model selection.

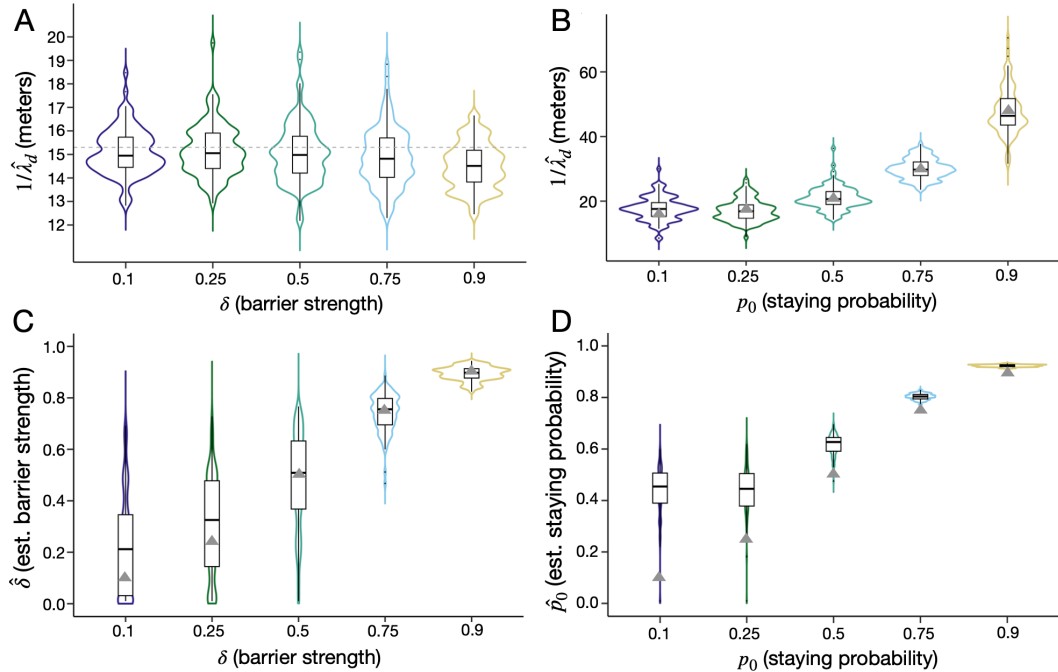

**Fig 6**. **Estimates of barrier strength and daily staying probability using spatial CKMR methods.** In the first (left) analysis, violin plots depict estimates of mean daily dispersal distance, $1/\lambda_d$ **(A)**, and barrier strength, $\delta$ **(C)**, obtained using the spatial CKMR approach for the optimal sampling scheme determined in Sect 3.1, and considering a barrier to movement as depicted in Fig 1D whereby movement to the other side of the barrier is reduced by a factor, $\delta$, in the range [0,0.9]. In the second (right) analysis, violin plots depict estimates of mean daily dispersal distance conditional upon movement, $1/\lambda_c$ **(B)**, and daily staying probability, $p_0$ **(D)**, in the range [0.1,0.9], again obtained using the spatial CKMR approach for the optimal sampling scheme determined in Sect 3.1, and considering a zero-inflated dispersal kernel as described in Eq 2 and depicted in Fig 1B. The simulated metapopulation consists of a 19-by-19 grid of households each inhabited by 25 adult *Ae. aegypti* at equilibrium with bionomic parameters listed in Table 1. Boxes depict median and interquartile ranges of 100 simulation-and-analysis replicates for each scenario, thin lines represent 5% and 95% quantiles, points represent outliers, and kernel density plots are superimposed. True parameter values are depicted by triangles, or by a dotted line when they are consistent across all cases.

## 4 Discussion

We have demonstrated that the CKMR formalism with spatial structure described by Bravington et al. [9] can be used to estimate dispersal parameters of *Ae. aegypti*, a major vector of arboviruses such as dengue and Zika virus, as a case study. Using a spatial individual-based simulation framework [18] based on the lumped age-class model [19] applied to mosquitoes [21], we have shown that these methods accurately estimate mean daily dispersal distance, $1/\lambda_d$, and can also estimate parameters of more complicated dispersal kernels, such as the strength of a barrier to movement, $\delta$, and the daily staying probability, $p_0$, for a zero-inflated exponential kernel. The optimal sampling scheme inferred in this study is consistent with standard *Ae. aegypti* field sampling protocols, provided the distribution of traps is carefully selected. As for a previous *in silico* mosquito CKMR analysis [18], we found that sampling adult females biweekly over a period of three months is adequate, as is commonplace for mosquito sampling [36] and consistent with the length of a season.

The simulated 19-by-19 grid landscape with nodes (in this case, households) spaced 16.6 meters apart was designed to resemble a suburban setting such as that of Cairns, Australia, as a case study (the neighborhood dimensions were chosen to resemble those of the Cairns suburb, Yorkeys Knob). In this setting, the optimal explored sampling scheme consisted of a five-by-five grid of trap nodes with two household nodes separating each trap node; however, a subsequent analysis on a larger landscape found the same grid of trap nodes with up to six or seven household nodes

separating each trap node also provided good inference. Of note, this equates to traps being separated by approximately the simulated mean lifetime dispersal distance (i.e., a separation distance of 49.8 meters c.f. a lifetime dispersal distance of 45.2 meters [11]) for the original analysis, and by up to 2.5 times the simulated mean lifetime dispersal distance for the subsequent analysis. This optimal trap layout and the optimal sample size of 2,500 adult females is amenable to *Ae. aegypti* field protocols. Dividing this total sample size across the full network of traps and a three-month sampling period equates to ca. 3-4 adult females sampled twice-weekly for each trap node. This sample size is achievable; but were it to present a challenge, then additional traps could be placed throughout the landscape. It is worth noting that smaller sample sizes would likely be required for smaller populations (our simulations assumed 25 adult mosquitoes per household [26]). Precise sample size requirements should ideally be inferred by repeating simulation and analysis replicates for the species and landscape of interest.

This formal CKMR approach to estimating mosquito dispersal parameters is complementary to the recently-proposed methods of Filipović et al. [11] and Jasper et al. [10], with each method having its own strengths and weaknesses. The Filipović et al. method is described in full in the Methods section of [11]; but in brief, adult females are captured while ovipositing, kinship categories determined, and a mean generational displacement calculated for each close-kin pair. This calculation considers the accumulated displacement between the close-kin individuals, and the set of possible movement events that led to it. A dispersal kernel is fitted to the set of mean generational displacements for all close-kin pairs. A slightly revised version of this method has been used by Ontiveros et al. [12]. The Jasper et al. method is described in full in the Materials and Methods section of [10]; but in brief, eggs are collected from ovitraps, kinship categories determined, and "axial standard deviations" are calculated for each kinship category. The mean dispersal distance of adult females between emergence and egg-laying is then calculated from variance formulae that incorporate the axial standard deviations of observed kinship categories (see Eqs 1–3 of [10]). This method can also be applied to sampled adult females, as demonstrated in [37], although it performs better when applied to sampled eggs.

Key strengths of the Jasper et al. [10] and Filipović et al. [11] methods are that: i) they are simpler than the formal CKMR approach, and hence require less computational investment, and ii) they accommodate second and third-degree close-kin without computational burden. This, however, makes a systematic comparison of the performance of the three methods difficult, as the Jasper et al. and Filipović et al. methods are ideally performed on larger landscapes that accommodate displacement accumulated over 2-3 generations, while the formal CKMR approach can be implemented on a smaller landscape, inferring mean daily dispersal from displacement accumulated over shorter time periods spanning one day through two generations. Of note, the Filipović et al. method can be applied exclusively to first-degree close-kin, making a reduced landscape amenable to that approach; however, this comes at the expense of sacrificing second and third-degree close-kin data that the method otherwise benefits from. For the formal CKMR approach, the incorporation of temporal information enables higher-resolution inference of dispersal parameters from data collected over a smaller area; however, the computational burden associated with this additional detail limits the size of a landscape that can be considered using this method.

A key benefit of the formal CKMR approach is its ability to estimate parameters of more complex dispersal kernels, such as the staying probability of a zero-inflated exponential kernel, and of more complex landscapes, such as the strength of a barrier to dispersal. Exponential and zero-inflated exponential kernels were explored in this analysis; but any number of dispersal kernels could be explored, provided available data is consistent with identifiability of their parameters. A related benefit of the CKMR approach is that it can be tailored to the life history and landscape of a specific mosquito species and location. This includes three-dimensional landscapes such as the multi-storey housing blocks analyzed in Singapore by Filipović et al. [11]. It should be noted that, with this capability comes a need to understand the local ecology of the species before parameter inference begins. E.g., when estimating dispersal parameters in this study, we incorporated a fully-specified life history model in addition to knowledge of the distribution of habitat patches and the functional form of the dispersal kernel. A sensitivity analysis on life history and other parameters at the simulation stage

could determine impacts on the precision and bias of dispersal parameters in the event that other parameters are poorly characterized. That said; the formal CKMR approach can also be used to estimate demographic parameters unrelated to dispersal, such as census population size, adult and larval daily mortality rates, and larval life stage duration, as shown in Sharma et al. [18], which could themselves inform spatial CKMR analyses to estimate dispersal parameters.

As a preliminary exploration of the application of formal CKMR methods to estimate dispersal parameters of mosquitoes, this study has several limitations. First, the same life history and landscape model (Fig 1) was used as a basis for both the population simulations and CKMR analysis. Additionally, other than the parameters being estimated, the same parameter values were used in both the simulations and analysis. This represents an overly generous scenario as compared to the field, where life history is varied and complex and parameters are only approximately known. The grid landscape, with each population having the same equilibrium population size, also represents an overly simplified scenario that does not capture the heterogeneity of real landscapes. That said; this is an appropriate starting point to verify the utility of the method, beyond which further research may be conducted. Second, we have assumed perfect kinship inference in our CKMR likelihood equations. Incorporating kinship uncertainty into the likelihood equations is theoretically possible [38], although this has produced little improvement in parameter inference at large computational cost when applied to data from fish species [32]. Fortunately, through introducing kinship assignment errors at the simulation phase, we have shown that inference of mean dispersal distance is robust to type II (false negative) errors, and can accommodate type I (false positive) errors of up to 5%. This suggests that, where trade-offs in kinship assignment are possible, it is best to err towards false negatives rather than false positives, in agreement with CKMR studies in fish species [9].

The application of formal CKMR methods to spatial settings, while contemplated since their inception [9], has only been considered in a small number of cases for species with simpler life histories [15,16]. The extension of these methods to a metapopulation of insects having an egg-larva-pupa-adult life history is promising for insights this approach may provide for other species, alongside insights from simpler close-kin-based approaches [10,11]. Potential applications to *An. gambiae* are of particular interest, given the importance of this species for malaria transmission and the importance of quantifying dispersal patterns for planning vector control interventions and field trials. The increased dispersal range of this species is important to acknowledge [5], as is the potential for age-grading techniques [39] to enhance parameter inference. Several species of insect agricultural pests share a similar life history, and close-kin-based approaches such as CKMR should be explored to provide insight into their dispersal patterns.

## 5 Conclusions

We have theoretically demonstrated the application of spatial CKMR methods to estimate dispersal parameters for mosquitoes, with *Ae. aegypti* as a case study. Close-kin-based methods have advantages over traditional MRR methods, as the mark is genetic, removing the need for physical marking and recapturing. Encouragingly, we find that optimal spatial CKMR sampling schemes are consistent with *Ae. aegypti* ecology and field studies, provided the spatial distribution of traps is carefully chosen. In our *in silico* case study, we found that traps distributed in a grid layout and separated by a distance of approximately 1–2.5 times the mean lifetime dispersal distance of this species were optimal. The formal CKMR approach is complementary to two simpler close-kin-based methods at estimating mean dispersal distance, with each approach having its own strengths and weaknesses. The CKMR approach is particularly computationally intensive, restricting its ability to be applied to second and third-degree close-kin and to larger landscapes; but its ability to be tailored to a specific landscape and dispersal kernel enable it to estimate additional parameters such as the daily staying probability and strength of a barrier to movement. Close-kin-based methods such as CKMR promise to provide further insight into the dispersal patterns of other insects of epidemiological and agricultural significance.

## Supporting information

**S1 Text. Supplemental model equations and results.** Additional equations describing mosquito dispersal dynamics, and spatial kinship probabilities for parent-offspring and full-sibling pairs that, for brevity, were not included in the manuscript. Select results are also included.
(PDF)

## Acknowledgments

We thank Dr. Yogita Sharma, Dr. Rachel Fewster and Dr. Mark Bravington for discussions regarding the application of CKMR methods to mosquito populations.

## Author contributions

**Conceptualization:** John M. Marshall, Gordana Rašić.

**Data curation:** John M. Marshall, Shuyi Yang, Jared B. Bennett, Igor Filipović, Gordana Rašić.

**Formal analysis:** John M. Marshall, Shuyi Yang, Jared B. Bennett, Igor Filipović, Gordana Rašić.

**Funding acquisition:** John M. Marshall, Gordana Rašić.

**Investigation:** John M. Marshall, Shuyi Yang, Igor Filipović, Gordana Rašić.

**Methodology:** John M. Marshall, Shuyi Yang, Jared B. Bennett, Igor Filipović, Gordana Rašić.

**Project administration:** John M. Marshall.

**Resources:** John M. Marshall, Gordana Rašić.

**Software:** John M. Marshall, Shuyi Yang, Jared B. Bennett.

**Supervision:** John M. Marshall.

**Validation:** John M. Marshall.

**Visualization:** John M. Marshall, Shuyi Yang, Gordana Rašić.

**Writing – original draft:** John M. Marshall.

**Writing – review & editing:** John M. Marshall, Shuyi Yang, Jared B. Bennett, Igor Filipović, Gordana Rašić.

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
