## [Decision Letter · Decision Letter 0]

13 Aug 2025

PCOMPBIOL-D-25-00982

Spatial close-kin mark-recapture methods to estimate dispersal parameters and barrier strength for mosquitoes

PLOS Computational Biology

Dear Dr. Marshall,

Thank you for submitting your manuscript to PLOS Computational Biology. After careful consideration, we feel that it has merit but does not fully meet PLOS Computational Biology's publication criteria as it currently stands. Therefore, we invite you to submit a revised version of the manuscript that addresses the points raised during the review process.

Please submit your revised manuscript within 60 days Oct 13 2025 11:59PM. If you will need more time than this to complete your revisions, please reply to this message or contact the journal office at ploscompbiol@plos.org. Please include the following items when submitting your revised manuscript:

We look forward to receiving your revised manuscript.

Kind regards,

Tyler Cassidy

Academic Editor

PLOS Computational Biology

Tobias Bollenbach

Section Editor

PLOS Computational Biology

**Additional Editor Comments:**

This is an interesting and well-presented manuscript. In general, the reviewers appreciated the approach and using CKRM to understand mosquito dispersal. However, both Reviewer 1 and 3 both noted figure 4 as being difficult to interpret (and not colourblind friendly). Further, Reviewer 3's comments on identifiability of model parameters and computational cost are important to address, given the presentation of the pseudo-likelihood in Section 2.3, and Reviewer 1's comments regarding Lines 493-498 and Lines 545-547. Finally, while many of the comments are minor, all three reviewers have made suggestions regarding additional analysis/situations. In the revision, please consider performing the additional analyses that may highlight the limitations or feasibility of the CKRM approach to be used in practice.

**Journal Requirements:**

<carina-action-element class="ng-star-inserted">2) Please upload all main figures as separate Figure files in .tif or .eps format. For more information about how to convert and format your figure files please see our guidelines: </carina-action-element><carina-action-element class="ng-star-inserted">https://journals.plos.org/</carina-action-element><carina-action-element class="ng-star-inserted">ploscompbiol</carina-action-element><carina-action-element class="ng-star-inserted">/s/figures</carina-action-element> 

**Reviewers' comments:**

Reviewer's Responses to Questions

**Comments to the Authors:**

Reviewer #1: Please see attached pdf

Reviewer #2: Uploaded as an attachment

Reviewer #3: Please see attached.

**Have the authors made all data and (if applicable) computational code underlying the findings in their manuscript fully available?**

Reviewer #1: Yes

Reviewer #2: Yes

Reviewer #3: Yes

PLOS authors have the option to publish the peer review history of their article (what does this mean?). If published, this will include your full peer review and any attached files.

Reviewer #1: **Yes: **Anthony Sévêque

Reviewer #2: No

Reviewer #3: No

**Figure resubmission:**
---

## [Editor Report · Decision Letter 1]

6 Nov 2025

Dear %TITLE% Marshall,

We are pleased to inform you that your manuscript 'Spatial close-kin mark-recapture methods to estimate dispersal parameters and barrier strength for mosquitoes' has been provisionally accepted for publication in PLOS Computational Biology.

Best regards,

Tyler Cassidy

Academic Editor

PLOS Computational Biology

Tobias Bollenbach

Section Editor

PLOS Computational Biology

I appreciate the thoughtful response to the reviewer comments and corresponding additions to the manuscript.

---

## [Editor Report · Acceptance letter]

PCOMPBIOL-D-25-00982R1

Spatial close-kin mark-recapture methods to estimate dispersal parameters and barrier strength for mosquitoes

Dear Dr Marshall,

I am pleased to inform you that your manuscript has been formally accepted for publication in PLOS Computational Biology. Your manuscript is now with our production department and you will be notified of the publication date in due course.

With kind regards,

Zsofia Freund
